# Intramolecular Hydrogen Bonds in Selected Aromatic Compounds: Recent Developments

**Aneta Jezierska [1], Peter M. Tolstoy [2] , Jarosław J. Panek [1] and Aleksander Filarowski [1,3,*]**

[1] Faculty of Chemistry, University of Wrocław, 14 F. Joliot-Curie str., 50-383 Wrocław, Poland; aneta.jezierska@chem.uni.wroc.pl (A.J.); jarek@elrond.chem.uni.wroc.pl (J.J.P.)

[2] Institute of Chemistry, St. Petersburg State University, Universitetskij pr. 26, 198504 St Petersburg, Russia; peter.tolstoy@spbu.ru

[3] Frank Laboratory of Neutron Physics, Joint Institute of Nuclear Research, 141980 Dubna, Russia

[*] Correspondence: aleksander.filarowski@chem.uni.wroc.pl; Tel.: +48-71-3757-229

**Abstract:** A review of intramolecular hydrogen bonding in *ortho*-hydroxyaryl *Schiff* bases, *ortho*-hydroxyaryl *Mannich* bases, dipyrrins, *ortho*-hydroxyaryl ketones, *ortho*-hydroxyaryl amides, and 4-Bora-3a,4a-diaza-s-indacene (BODIPY) dyes with tautomeric sensors as substituents is presented in this paper. *Ortho*-hydroxy *Schiff* and *Mannich* base derivatives are known as model molecules for analysing the properties of intramolecular hydrogen bonding. The compounds under discussion possess physicochemical features modulated by the presence of strong intramolecular hydrogen bonds. The equilibrium between intra- and inter-molecular hydrogen bonds in BODIPY is discussed. Therefore, the summary can serve as a knowledge compendium of the influence of the hydrogen bond on the molecular properties of aromatic compounds.

**Keywords:** *Schiff* base; cyclically arranged hydrogen bond; acetophenone; amide; BODIPY

## 1. Introduction

This review focuses on selected studies of so-called *chelate* formations. Investigations of *chelate* formations possessing either hydrogen or metal atoms, "caught by *chelate*", have attracted much interest of chemists and biochemists [1–5]. It is necessary to stress that studies of *chelate* formations are important in the modeling of new catalysts [6–10]. In Refs. [11–15], careful analyses of the impact of hydrogen bonding on catalytic reactions are presented. This review mostly discusses studies of compounds with hydrogen bonding. The last part of the review covers studies of the influence of hydrogen bonding of the *chelate* type on the N-B(F$_2$)-N formation of the 4-Bora-3a,4a-diaza-s-indacene (BODIPY) dye.

In this review, the described compounds feature intramolecular hydrogen bonding; some of the discussed molecules possess so-called *quasi*-aromatic hydrogen bonds [16–18]. The studied intramolecular hydrogen bonds can be classified into two groups, according to their strengthening effects: *Resonance Assisted Hydrogen Bonds* (RAHBs) [19] and *Charge Assisted Hydrogen Bonds* (CAHBs) [20,21] or ionic hydrogen bonds [22]. As the objects for this review, we have selected *ortho*-hydroxyaryl and *ortho*-hydroxyalkyl *Schiff* bases, *Mannich* bases, ketones, dipyrrins, and amides, as well as fluorescent BODIPY dyes with tautomeric sensors as substituents. The structures of the discussed compounds are shown in Figure 1. A majority of the presented studies have been published over the last decade and cover the phenomena of proton transfer (PT), the steric effect, quasi-aromatic hydrogen bonding, aromaticity, and tautomeric and conformational equilibria. It should be stressed that studies of the proton transfer process are of great importance in the description of some enzymatic reactions activated by hydrogen bonding with a low energy barrier for proton transfer (LBHB—low

barrier hydrogen bond) [23,24]. The studies of *ortho*-hydroxyaryl *Schiff* bases and hydrogen bonds can also contribute to understanding on the mechanisms of activity of some drugs, as well as support the design of new materials [25–27].

**Figure 1.** The structures of the hydroxyaryl *Schiff* bases (**I**), hydroxyalkyl *Schiff* bases (**II**), hydroxyaryl *Mannich* bases (**III**), dipyrromethene (**IV**), dipyrromethane (**V**), hydroxyl acetophenones (**VI**), hydroxyl benzamides (**VII**), and 4,4-difluoro-8-[4-(methoxycarbonyl)phenyl]-1,3,5,7-tetramethyl-3a,4a-diaza-4-bora-s-indacene (BODIPY) (**VIII**). R–R$_2$ refer to the aryl and alkyl substituents.

## 2. Impact of Resonance Assistance and Charge Assistance on the Hydrogen Bond Properties

The papers [28] and [29] dwell on calculations of the hydroxyaryl *Schiff* and *Mannich* bases by the Density Functional Theory (DFT) method, illustrating the influence of proton transfer in *quasi*-aromatic hydrogen bonding on the shape of the potential energy curve. Ref. [28] presents the use of the methodology of potential calculations for proton movement in the hydrogen bridge and for spectroscopic parameters. The following properties were calculated in this study: the non-adiabatic and adiabatic potentials, as well as the vibrational energy levels for the *ortho*-hydroxyaryl ketoimines and their deutero-derivatives. The spectroscopic parameters calculated from the energy levels allowed for an accurate assignment of the XH stretching band positions in experimental IR spectra. The presented calculations of the non-adiabatic potential for the proton transfer process are the result of the optimization of all parameters of the molecule for the various lengths of the XH bond. This approximation made it possible to accurately describe the changes of structural parameters under the transition from the *molecular* form to the *proton transfer* form. It is stated that proton transfer brings about the shortening and strengthening of the hydrogen bridge in the first phase of this process (the transition from the *molecular* form to the *transition* state), with the consequent lengthening and bending being viewed in the second phase (the *transition* from the intermediate state to the *proton transfer* form), as seen in Figure 2.

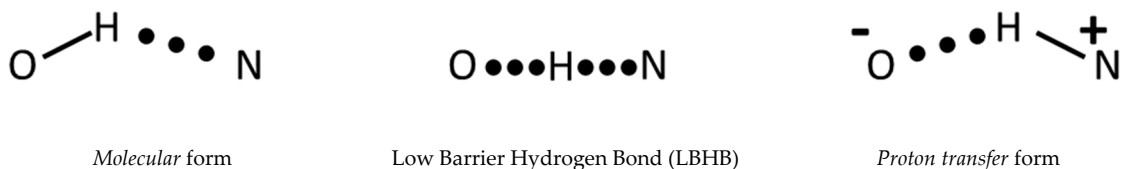

Molecular form　　　　　Low Barrier Hydrogen Bond (LBHB)　　　　　Proton transfer form

**Figure 2.** The principal structural stages of proton transfer in the hydrogen bridge.

For a description of the electron density changes occurring in the molecule under proton transfer in the hydrogen bridge, the method of "Atoms in Molecules" (AIM) [30] was employed. In Ref. [29], a complete analysis of the electron density changes at the bonds' and rings' critical points obtained by calculation of the potential energy curve for proton transfer within the adiabatic approximation for 2-((1-methylo-imino)ethyl)phenol (Figure 3) is described.

**Figure 3.** The structure of 2-((1-methylo-imino)ethyl)phenol.

Studies of the changes in tautomeric equilibrium in *ortho*-hydroxyaryl *Schiff* bases have mainly used two tautomeric forms (the *molecular* and *proton transfer* forms). However, the dependence of electron density changes at the critical point of the *chelate* ring ($\rho_{RCP}$) on the length of the XH bond (Figure 4) shows that an accurate description of the tautomeric equilibrium requires an intermediate point (transition state). In line with the studies of potentials, the transition state can be characterized by the most linear and shortest hydrogen bridge (LBHB). Refs. [29–37] focused on studies of *quasi*-aromatic hydrogen bonding. The formation of intramolecular *quasi*-aromatic hydrogen bonding via delocalization of the π-electrons in these compounds stabilizes a plain conformation of the molecule and strengthens the hydrogen bonding.

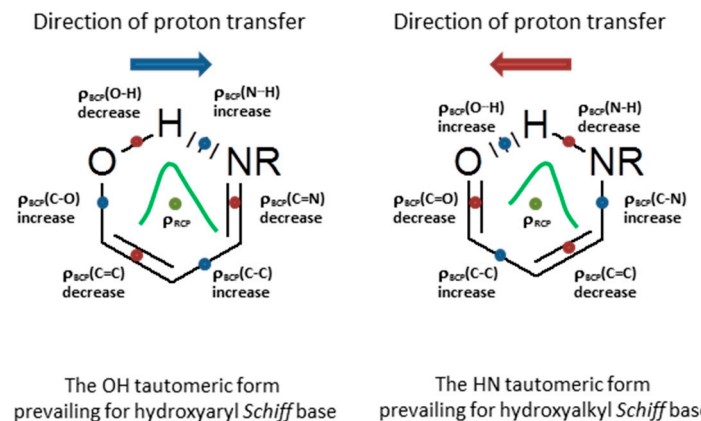

**Figure 4.** The scheme of electron-density topological changes in *quasi*-aromatic hydrogen bonding under the proton transfer process in the hydrogen bridge; on the left—for the compounds with the prevailing enol-imine form (the hydroxyaryl *Schiff* bases (**I**)), on the right—for those with the prevailing keto-amine form (the hydroxyalkyl *Schiff* bases (**II**)). Ref. [29]—Adapted by permission of The Royal Society of Chemistry.

Refs. [38–42] developed a description of the changes of molecules' aromaticity, depending on the state of the tautomeric equilibrium. X-ray data and quantum-mechanical calculations enabled the authors to obtain the dependencies of the Harmonic Oscillator Model of Aromaticity (HOMA) index on the XH bond length. The HOMA index was introduced and developed by Krygowski et al. [43]. It was also proved that the changes of aromaticity in the *chelate* chain (HOMA(ch)) non-linearly depend on the changes of aromaticity of this chain under proton transfer (HOMA(ch) = $f$(d(OH/HN)), Figure 5) [29]. Moreover, it was shown that changes in the aromaticity of the *chelate* chain non-linearly depend on the structural parameters of the hydrogen bridge under the proton transfer process (HOMA(ch) = $f$(d(OH/HN), Figure 5).

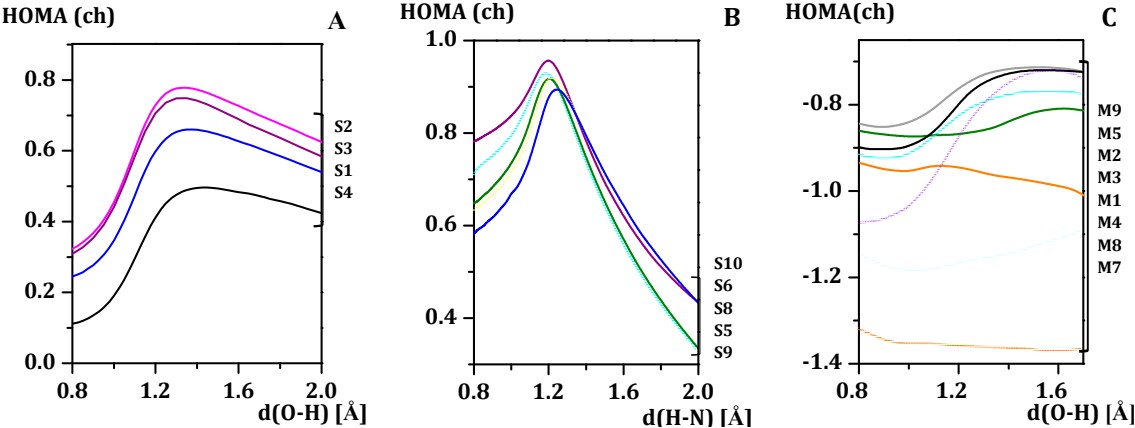

**Figure 5.** The dependencies of the aromaticity index of the *chelate* ring (HOMA(ch)) on the OH or HN bond lengths (d(OH/HN), in [Å]) for aryl (**I**, chart **A**) and alkyl *Schiff* bases (**II**, chart **B**), as well as *Mannich* bases (**III**, chart **C**). Ref. [29]—Reproduced by permission of The Royal Society of Chemistry.

A comparison of two families of compounds—with π-electronic coupling (hydroxyaryl and hydroxyalkyl *Schiff* bases) and without π-electronic coupling (hydroxyaryl *Mannich* bases)—verified the dependence of the hydrogen bonding energy on the degree of π-electronic coupling in the *chelate* chain. The HOMA aromaticity index served as an indicator of the π-electronic coupling degree. The comparison of two groups of compounds showed a significant difference between them. These studies demonstrated that the influence of π-electronic coupling in the *chelate* chain increases the energy of the intramolecular hydrogen bond by 30%. It is necessary to point out that this effect is a few times larger for the transition state, when π-electronic delocalization in the chelate chain is the maximum (Figure 6). These results, elucidated in publication [14], made it possible to find a tool for the analysis of RAHB, namely the dependence of the energy of hydrogen bonding on the HOMA aromaticity index of the *chelate* chain ($\Delta E_{HB} = f$(HOMA(ch)), Figure 6).

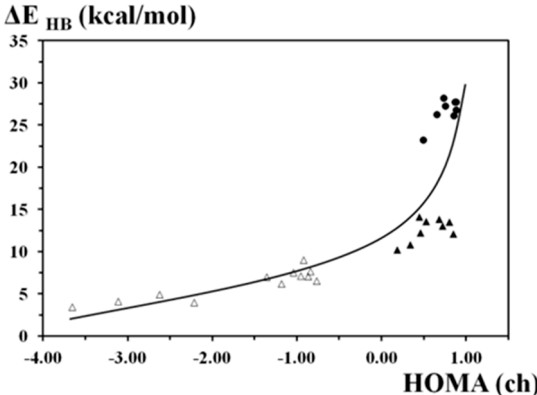

**Figure 6.** The dependence of the hydrogen bond energy on the aromaticity index of the *chelate* chain (HOMA(ch)). Dark triangles and circles refer to the tautomeric form OH and the transition state for the compounds with *quasi*-aromatic hydrogen bonding (*Schiff* bases); light triangles stand for the OH tautomeric form for the compounds without π-electronic coupling (*Mannich* bases). Ref. [29]—Reproduced by permission of The Royal Society of Chemistry.

The main objective of paper [44] was to study the compounds with resonance assisted hydrogen bonding (RAHB in dipyrromethene—**III**) and charge assisted hydrogen bonding (CAHB in dipyrromethane anion—**IV**). The dipyrrines studies performed proved that the CAHB is stronger than the RAHB in terms of the global energy minimum. The reason for this is that the strengthening of hydrogen bonding via the charge (CAHB) is more powerful than the strengthening by resonance

(RAHB). However, it is of importance that the increase of π-electronic coupling (the growth of the HOMA(ch) index) causes the strengthening of hydrogen bonding in dipyrromethene if compared with the hydrogen bonding in the dipyrromethane anion. The process of proton transfer is accompanied by shortening of the hydrogen bridge in dipyrromethene, as well as in the dipyrromethene anion. The lengths of the hydrogen bonds of both derivatives are quite similar in the transition state, whereas, in the global minimum state, the hydrogen bonding in the dipyrromethene is longer than for the dipyrromethane anion. These results support the following conclusion: the strengthening of the hydrogen bonding via π-electronic coupling in the *chelate* chain appears to be the most efficient in LBHB (Table 1).

**Table 1.** The values of energy for the hydrogen bonding in dipyrromethene (**III** and **III(TS)**—for the optimized geometries and transition states, respectively) and in the dipyrromethane anion (**IV**), calculated with the following equations: (**1**) $6.4 \times 10^3 \times e^{-3.1 \cdot d(HN)}$ (Å); (**2**) $3.8 \times 10^3 \times e^{-2.73 \times d(HN)}$ (Å); (**3**) $-5.554 \times 10^5 \times e^{-4.12 \times d(NN)}$ (Å); (**4**) $186 \times \rho_b - 2.3$ (eÅ$^3$); (**5**) $163 \times \rho_b$ (eÅ$^3$); (**6**) $0.448 \times G_b - 3.1$ (kcal·mol$^{-1}$·a$_0$$^{-3}$); (**7**) $0.429 \times G_b$ (kcal·mol$^{-1}$·a$_0$$^{-3}$); (**8**) $-0.37 \times V_b + 3.1$ (kcal·mol$^{-1}$·a$_0$$^{-3}$); (**9**) $-0.31 \times V_b$ (kcal·mol$^{-1}$·a$_0$$^{-3}$); and (**10**) $0.5 \times V_b$ (kcal·mol$^{-1}$·a$_0$$^{-3}$) [the references are in Ref. [45]]. The values d(N H), d(N N), $\rho_b$, $G_b$, and $V_b$ were obtained by an ab-initio method (MP2/6-311++G(2df,2pd)). Data adapted from Ref. [44]. Copyright 2014, The American Chemical Society.

| Cmpd. | Method\Equation | 1 | 2 | 3 | 4 | 5 | 6 | 7 | 8 | 9 | 10 |
|---|---|---|---|---|---|---|---|---|---|---|---|
| **III** | MP2/6-311++G(2df,2pd) | 3.30 | 4.20 | 7.35 | 5.37 | 7.96 | 5.77 | 6.25 | 6.29 | 4.63 | 7.45 |
| **III(TS)** | MP2/6-311++G(2df,2pd) | 28.19 | 27.55 | 23.17 | 43.61 | 41.47 | 18.19 | 18.14 | 49.59 | 40.93 | 65.81 |
| **IV** | MP2/6-311++G(2df,2pd) | 5.53 | 6.59 | 9.43 | 9.47 | 11.56 | 7.99 | 8.37 | 9.95 | 7.70 | 12.39 |

The presented results state the following conclusions: (1) the comparison of *ortho*-hydroxyaryl *Schiff* bases and *Mannich* bases confirms that resonance assistance greatly strengthens the hydrogen bond (by approximately 30%), and (2) the comparison of dipyrromethene and dipyrromethane shows that assistance by charge strengthens the hydrogen bond more effectively than resonance assistance. However, the RAHB strength is similar to the CAHB strength in the transition state.

## 3. Quasi-Aromatic Cyclically Arranged Hydrogen Bonds

Refs. [46,47] present analyses of compounds with hydrogen bonds in the form of a circle (Figure 7). They show that the steric squeezing of cyclic *quasi*-aromatic hydrogen bonds brings about a drastic shortening of the hydrogen bridge in *tris*-hydroxy ketoimines.

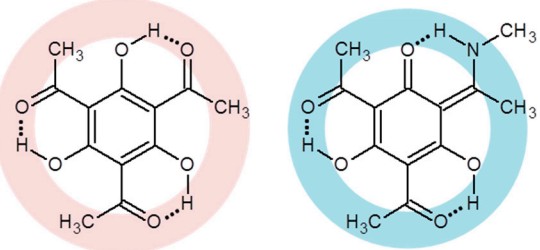

**Figure 7.** The derivatives of the phloroglucin (1,1′,1″-(2,4,6-trihydroxybenzene-1,3,5-triyl)trietanon, left structure) and *tris*-hydroxy ketoimine structures ((1E)-1-(3,5-diacetyl-2,4-dihydroxy-6-oxocyclohexane-2,4-dieno-1-yl)-N-methylethaniminium, right structure).

In Refs. [46–51], the concept of conformational equilibrium and dynamics of the bridging protons (amplitude of the movement of hydrogen) in the cyclically arranged hydrogen bonds was developed. According to the results obtained by the Car–Parrinello molecular dynamics (CPMD) method, the dynamics of the hydrogen bridge (changes of the distance between the proton-donating and proton-accepting atoms) increases with the shortening of the hydrogen bond. This occurs until the equilibrium proton position is located at the donor side. When the equilibrium is shifted towards the proton-transferred tautomer, the dynamics of the bridging proton usually weaken. However, strong correlations of proton motions in the triple cycle were not found. The paper [46] concentrated on experimental and theoretical studies of vibrations of the hydrogen bridge in the far infrared spectral range. The spectral measurements were performed using inelastic incoherent neutron scattering (IINS), infrared (IR) testing, and Raman spectroscopy in the wide spectral and temperature range. The assignments of bands corresponding to vibrations of the hydrogen bridge were based on deutero-substitution and an analysis of quantum-mechanical calculations at Density Functional Perturbation Theory (DFPT) and DFT levels. It was shown that the DFPT calculation method most accurately describes the experimental spectra.

According to the measured vibrational spectra (IR, Raman, and IINS, Figure 8), it can be noted that in the IINS spectrum, there are intense bands of torsion vibrations of methyl groups which are not seen in the IR and Raman spectra. The spectroscopic studies performed for the cyclically arranged hydrogen bridge systems proved that the IINS method provides the most accurate results in the far infrared region.

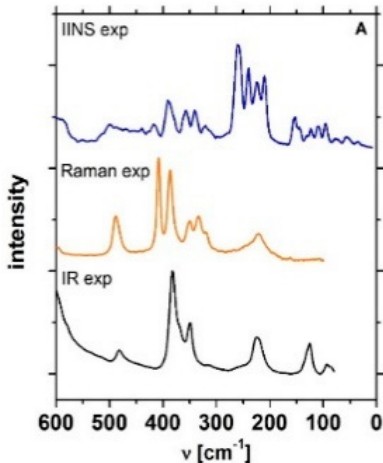

**Figure 8.** The experimental infrared (IR), Raman, and inelastic incoherent neutron scattering (IINS) spectra of 1,1′,1″-(2,4,6-trihydroxybenzene-1,3,5-triyl)trietanon measured in the solid state. Reproduced from Ref. [46] with permission from the Centre National de la Recherche Scientifique (CNRS) and The Royal Society of Chemistry.

The advantage of the IINS method is that the neutron scattering is only sensitive to vibrations of the bridging hydrogen. In the paper [46], the $\gamma(XH) = f(R(XY))$ correlation (earlier published by Tomkinson et al. [52,53]) was verified and the non-linear character of this correlation for intra- and inter-molecular hydrogen bonds was confirmed. The spectroscopic studies allowed one to gain fundamental knowledge on hydrogen bridge modes, unambiguously interpreting the phenomenon of polymorphism in 5-chloro-3-nitro-2-hydroxy-acetophenone (Figure 9).

**Figure 9.** The structure of 5-chloro-3-nitro-2-hydroxyacetophenone (**VI**).

In the studies of polymorphism in 5-chloro-3-nitro-2-hydroxy-acetophenone [54] and cyclically arranged hydrogen bonds in *tris* ketoimines, vibrations of the intramolecular hydrogen bridge in the far vibrational range were also examined (Figure 10). It was demonstrated that the hydrogen bonding vibrations can be divided into symmetric and asymmetric stretching vibrations and the deformation vibration. The studies of Far-IR, Raman, and IINS spectra stated that the symmetric and asymmetric stretching vibrations are combinations of vibrational modes of the bridging hydrogen, as well as the proton-donor and proton-acceptor groups. Meanwhile, the deformation vibrations were not necessarily related to the bridging hydrogen vibrations.

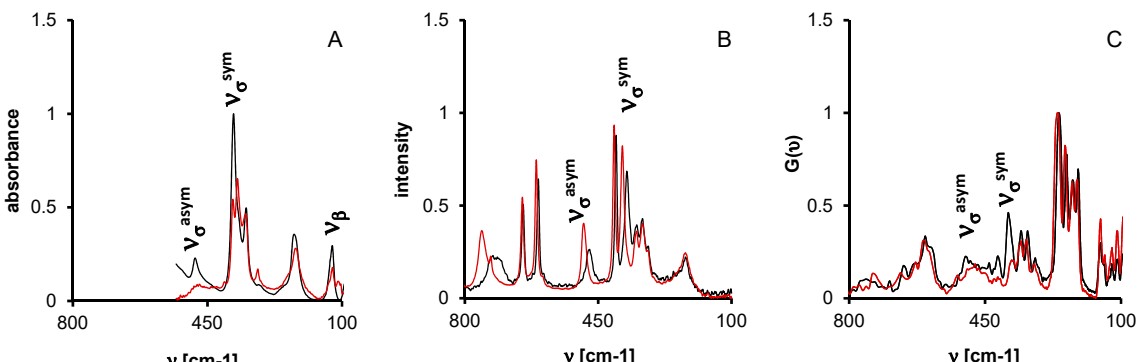

**Figure 10.** The infrared spectra (absorbance, **A**), Raman (intensity, **B**), and IINS (*G(v)*, **C**) of 1,1′,1″-(2,4,6-trihydroxybenzyl-1,3,5-triyl)triethanone (black line) and their deutero-derivatives (red line). Ref. [46] – Adapted by permission of The Royal Society of Chemistry (RSC) on behalf of the Centre National de la Recherche Scientifique (CNRS) and the RSC.

Further research into the issue of competitive hydrogen bridge interactions in structurally similar compounds—5-chloro-3-nitro-2-hydroxy-acetophenone [54] and 5-methyl-3-nitro-2-hydroxy-acetophenone [55,56]—exposed the existence of a competitive equilibrium in matrix conditions and the phenomenon of polymorphism in the solid state (Figure 11).

**Figure 11.** The scheme of the conformational equilibrium of 5-R-3-nitro-2-hydroxy-acetophenone, where R = Cl or $CH_3$.

To explain the nature of these two phenomena, the following methodology was used in Ref [54]: 1. The initial calculations of potential energy curves for rotation of the nitro group and simultaneous rotation of the hydroxyl and acetyl groups were performed to determine the stable and metastable states; 2. two polymorphs of the compound were obtained by crystallization from chloroform and methanol, respectively; 3. the crystal structures of both polymorphs were refined by the X-ray method (Figure 12); 4. the phase transition in one of the polymorphs was revealed by the Differential Scanning Calorimetry DSC method, which was confirmed by a $^{35}$Cl Nuclear Quadrupole Resonance (NQR) study (Figure 13); 5. the IR, Raman, and IINS spectroscopic studies revealed the bands most sensitive to the isomeric balance (in the matrix condition) and polymorphic changes in the solid state; 6. the method of isotopic substitution in the hydrogen bridge (OH→OD) was used for the bands' assignment. The methodology grounded on quantum-mechanical calculations gives both the possibility of the examination of these phenomena in different environments and the description of the observed physicochemical phenomena.

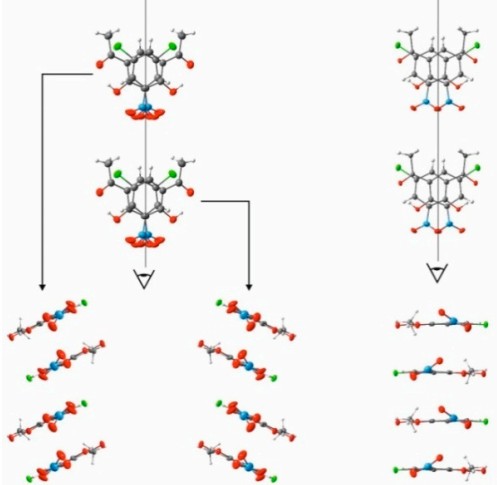

**Figure 12.** The fragments of packing of two polymorphic states of 5-chloro-3-nitro-2-hydroxy-acetophenone. Reprinted from Ref. [54]. Copyright 2019, The American Chemical Society.

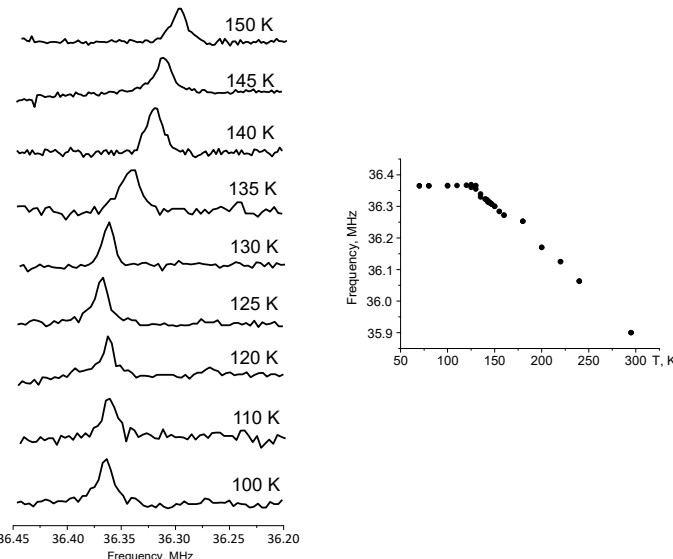

**Figure 13.** The $^{35}$Cl Nuclear Quadrupole Resonance (NQR) spectra of 5-chloro-3-nitro-2-hydroxy-acetophenone and the plot of $^{35}$Cl NQR resonance frequency as a function of temperature. Reprinted from Ref. [54]. Copyright 2019, The American Chemical Society.

## 4. Equilibrium between Intramolecular and Intermolecular Hydrogen Bonds

The compounds containing the amide moiety were examined by different techniques in Refs. [57–60]. The amide group is one of the most important structural motifs and it appears to be crucial in the functioning of living organisms [61–65]. Therefore, a detailed description of the conformation of systems with the amide moiety is very important. In the papers [66–69], it was shown that salicylamides can form two types of hydrogen bonds—intramolecular in 5-chloro-2-hydroxy-benzamide and intermolecular in 2-hydroxy-N,N-diethylbenzamide in the solid state (Figure 14).

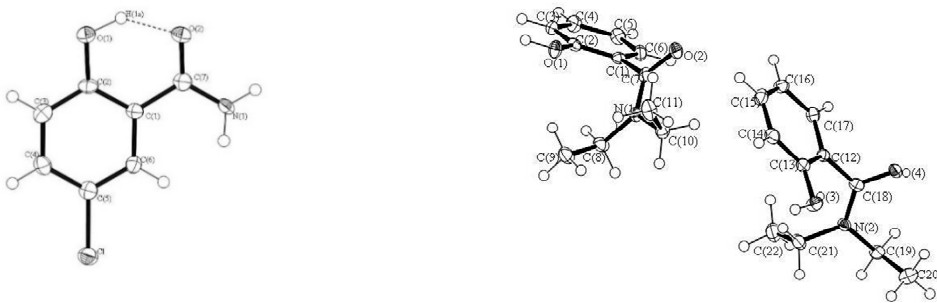

**Figure 14.** The single crystal structures of 5-chloro-2-hydroxy-benzamide and 2-hydroxy-N, N-diethylbenzamide.

Studies of the IR, NMR (Nuclear Magnetic Resonance), and UV-Vis spectra, recorded as a function of temperature in non-polar, proton-accepting, and proton-donating solvents [68,69], prove the existence of an equilibrium between intramolecular and intermolecular hydrogen bonds. It is noteworthy that the ease of breaking the strong intramolecular hydrogen bond (e.g., for 4-chloro-2-hydroxybenzamide $d(O\cdots O) = 2.526$ Å) in 2-hydroxy-N,N-diethylbenzamide is the result of a strong steric effect of ethyl groups and weakened π-electronic coupling between the phenyl and amide fragments [67]. In Ref. [68], the equilibrium between the intermolecular and intramolecular hydrogen bonds, as well as the rotation of the dimethylamine, morpholine, or pyrrolidine groups, were analysed.

In Ref. [68], the [1]H and [13]C NMR spectra of 2-hydroxy-N,N-diethylbenzamide, measured as a function of temperature and solvent polarity, showed the existence of a dynamic effect resulting from the orientation of ethyl groups. On the basis of the temperature-dependent [1]H and [13]C NMR spectra (Figure 15) and DFT calculations, it was found that the intramolecular hydrogen bond can be observed in aprotic solvents for β-hydroxy naphthalene amides, whereas, in protic solvents (based on NMR data) and in the solid state (based on X-ray data), the intermolecular hydrogen bond is present. The energy barrier for the rotation around the C-N bond was calculated and equaled 13–19 kcal/mol for β-hydroxy naphthalene amides dissolved in aprotic solvents such as chloroform, acetone, or tetrahydrofuran. The heights of the energy barriers correlated with the hydrogen bond strength ([1]H NMR chemical shift of the hydroxyl group was used as a measure of the hydrogen bond energy), as seen in Figure 16.

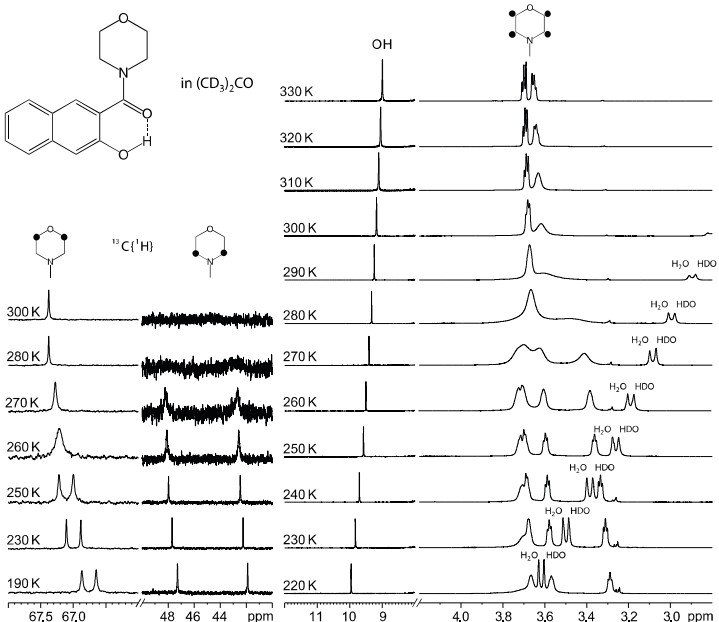

**Figure 15.** The $^1$H and $^{13}$C NMR spectra β-hydroxy naphthalene amide derivative as a function of temperature in (CD$_3$)$_2$CO. Reprinted from Ref [68]. Copyright 2015, Elsevier.

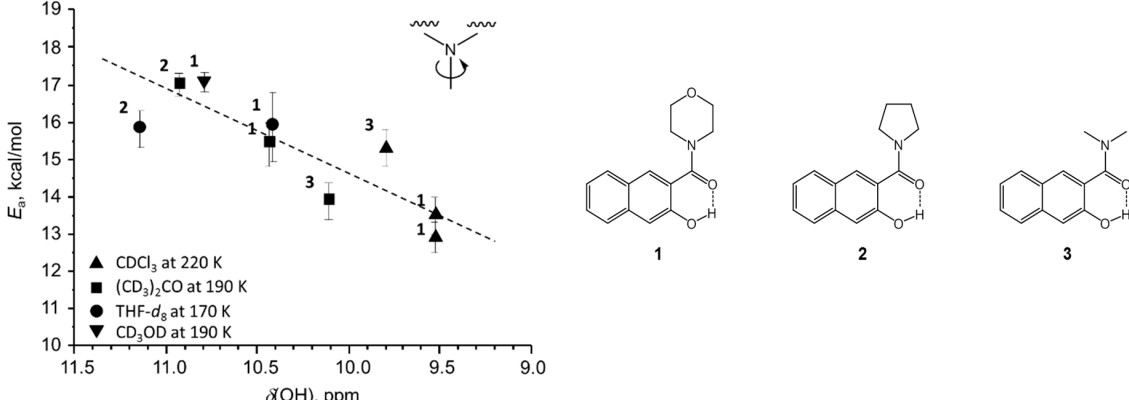

**Figure 16.** The dependence of the energy barrier ($E_a$) of the rotation around the C-N amide bond from Table 1. NMR chemical shift of the hydroxyl group for (3-hydroxynaphthalen-2-yl), (morpholin-4-yl)methanone (**1**), (3-hydroxynaphthalen-2-yl)(pyrrolidine-1-yl)methanone (**2**), and 3-hydroxy-N,N-dimethylnaphthalene-2-carboxamide (**3**), dissolved in the solvents listed in the plot. Reprinted from Ref. [68]. Copyright 2015, Elsevier.

## 5. Effect of Substituents and Solvatochromism of the Fluorescent 4-Bora-3a,4a-diaza-s-indacene Chromophore (BODIPY)

The growing popularity of fluorescent dyes rests upon the possibility of chemical modification of both the chromophore core and its substituents, which can manifest in a variety of spectroscopic properties [70–83]. The main objective of fluorescent BODIPY (4,4′-difluoro-4-boron-3a,4a-diaza-s-indacene) dye studies has been to explore the influence of the tautomeric equilibrium sensor on the spectroscopic parameters of the dye [84–88]. Below, we present a summary of our studies on BODIPY dyes with tautomeric sensors. The structures of the discussed compounds are shown in Figure 17.

**Figure 17.** The structures of 4,4-difluoro-8-[4-(methoxycarbonyl)phenyl]-1,3,5,7-tetramethyl-3a,
4a-diaza-4-bora-s-indacene (BODIPY, (**VIII**)), 4,4-difluoro-3-{2-[4-(dimethylamino)phenyl]ethenyl}-8-
[4-(methoxycarbonyl)-phenyl]-1,5,7-trimethyl-3a,4a-diaza-4-bora-s-indacene (dimethylamino) phenyl-
BODIPY (**IX**), BODIPY-*Schiff* base (**X**), and BODIPY-diketone (**XI**).

The results of the studies on strong fluorescent BODIPY dyes, including their spectroscopic and
photophysical properties depending on the Lippert-Mataga and $E^N_T(30)$ parameters, which describe
the effect of the solvent on the dye, were presented in Ref. [84]. Notably, not all the spectroscopic
correlation types (spectroscopic parameter = $f$(parameter determining the solvent impact)) can reflect
the physical phenomena occurring in the molecule at the transition from the ground state to the
electronic excited state. It was found that the spectroscopic parameters of compound **VIII** poorly
respond to the changes in solvent polarity. However, a significant bathochromic shift was observed for
compound **IX**, accompanied by an increase of the solvent polarity. It is noteworthy that the correlation
of the Stokes shift $\Delta\nu = f(E^N_T(30))$ exposes the change of spectral values when the N,N-dialkylamine of
compound **IX** interacts with protic solvents (Figure 18). The observed decrease of the Stokes shift can
be explained by the reduction of the charge transfer from the N,N-dialkylamine moiety to the BODIPY
core. The impact on the Stokes shift is a specific interaction, such as the intermolecular hydrogen bond
type, which significantly enhances the bathochromic shift of the emission band due to the weakening
of coupling between the N,N-dialkylamine moiety and the BODIPY core.

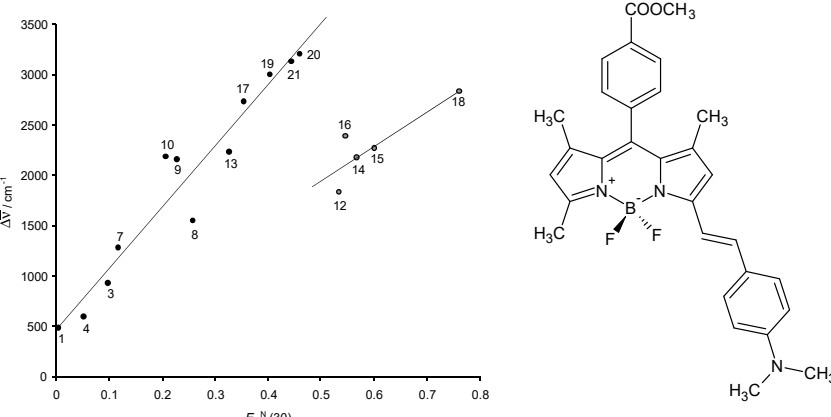

**Figure 18.** The dependence of the Stokes shift ($\Delta\bar{\nu}$) on the solvent polarity parameter $E^N_T(30)$ for **IX**.
The numbers 12, 14, 15, 16, and 18 correspond to the protic solvents, and all other numbers correspond
to the aprotic ones. Ref. [84]—Adapted by permission of The Royal Society of Chemistry (RSC) on
behalf of the European Society for Photobiology, the European Photochemistry Association, and RSC.

The subsequent studies of fluorescent dyes in Refs. [84–88] successfully used the spectral parameters' dependences on the parameters defining the polarity of the environment for the detection and description of processes occurring in the electronic ground and excited states. Refs. [87,88] present syntheses and studies of the BODIPY dyes with substituents sensitive to protonation and deprotonation, as well as to conformational equilibria in BODIPY-*Schiff* (**X**) and BODIPY-diketone dyes (**XI**), as seen in Figure 19. These studies were performed by experimental methods and DFT calculations in electronic ground and excited states. The idea of substitution at position 5 was based on the strongest electronic coupling occurring between the dye core and the substituent (diketone or *ortho*-hydroxyaryl *Schiff* base). These substituents feature *keto-enol* equilibrium (diketone moiety) and intramolecular proton transfer (*ortho*-hydroxy aldimine moiety), as well as isomerization around the imine bond conditioned by the polarity of the environment and specific interactions such as intermolecular hydrogen bonding (Figure 19).

**Figure 19.** The schemes of equilibria in the fluorescent BODIPY-*Schiff* and BODIPY-diketone dyes.

The fluorescent BODIPY-*Schiff* and BODIPY-diketone dyes were studied in a wide range of solvent polarities in electronic ground and excited states [87,88]. The effects of the solvent polarity and specific interactions on the keto-enol equilibrium, isomerization of the imine bond, and intramolecular versus intermolecular hydrogen bond equilibrium on the spectroscopic characteristics of the BODIPY-chromophore were investigated. Spectroscopic studies of the dyes showed a significant difference in the behavior of the absorption and emission bands in view of the specificity of the interactions. An increase of solvent polarity leads to a bathochromic shift of the absorption band for the BODIPY-*Schiff* dye, whereas such an increase for the BODIPY-diketone dye causes a hypsochromic shift. The emission bands' behavior also differs. In the proton-accepting solvents, the emission band is significantly bathochromically shifted for the BODIPY-*Schiff* dye, whereas for the BODIPY-diketone dye, the emission band is practically not shifted in the polar environment (Figure 20).

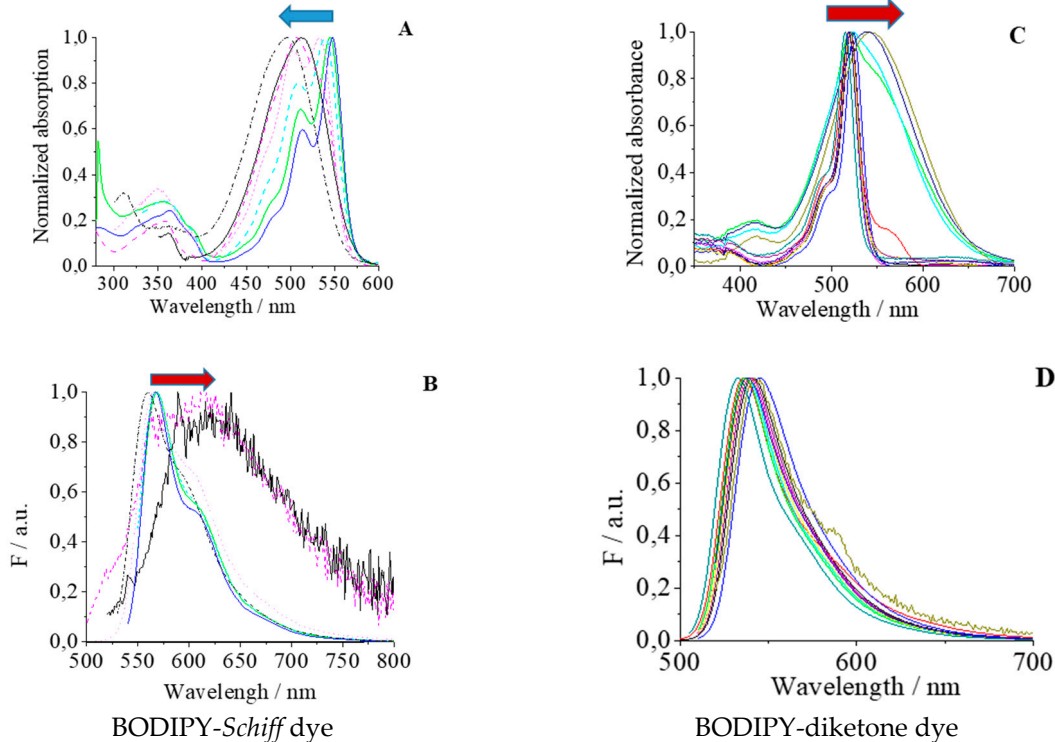

**Figure 20.** The selected absorption (**A**,**C**) and emission (**B**,**D**) spectra of BODIPY-*Schiff* and BODIPY-diketone dyes, respectively, in variations of solvents. The list of solvents is presented in Refs. [87,88]. Adapted from Refs. [87,88]. Copyright 2015 and 2018, The American Chemical Society.

An analysis of the impact of isomeric equilibria on the BODIPY-*Schiff* dye was performed on the basis of experimentally obtained correlations $\Delta\nu = f(\Delta f)$ and $\Delta\nu = f(E^N_T(30))$ (Figure 21).

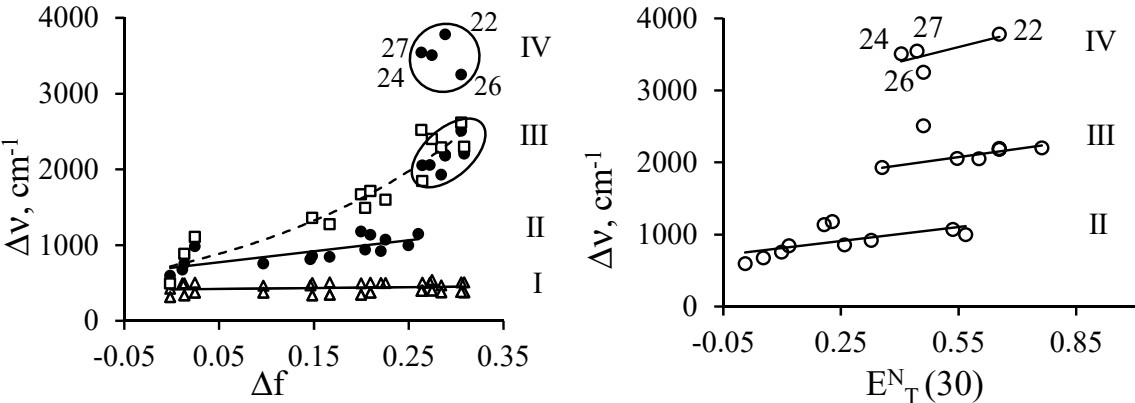

**Figure 21.** The dependencies of the Stokes shift ($\Delta\nu$) on Lippert-Mataga ($\Delta f = f(\varepsilon) - f(n^2)$) and $E^N_T(30)$ parameters for BODIPY-*Schiff* dye (curves II, III and scatterplot IV; black circles) and dyes **VIII–IX** (empty triangles and squares, curves I and III). The dependencies for solutions with a dielectric permittivity $\varepsilon < 17.8$ (I and II); $\varepsilon > 17.8$ (III); and DMSO, DMF (dimethylformamide), $C_2H_5OH$, + KOH (IV). Reprinted with permission from Ref. [87]. Copyright 2015, The American Chemical Society.

Correlation I (Figure 21) indicates that there are no significant structural changes under the excitation of dyes **VIII** and **IX**. Meanwhile, for the BODIPY-*Schiff* dye, at least two structural changes occur with an increasing solvent polarity, as well as due to specific interactions. The solvent polarity increase leads to proton transfer (black circles III) and isomerization in the proton-accepting solvents

(black circles IV). This effect is clearly visible for the correlations $\lambda_{abs/em} = f(\Delta f)$, $\lambda_{abs/em} = f(E_{N/T}\Delta\bar{\nu}$ (30)), $= f(\Delta f)$ and $= f(E_{N/T}(30))$, which can be treated as an indicator of the structural changes. Presumably, the structural changes can be described using two isomeric equilibria (Figure 19). To determine the conformational states and explain the observed phenomenon, $^{1}$H NMR spectra were also measured and DFT/TD-DFT calculations for the electronic ground and excited states were performed. The NMR spectra showed that the studied compound in the ground state only exhibits *keto-enol* equilibrium. The experimental results were confirmed by DFT and TD-DFT (Time-Dependent Density Functional Theory) calculations of the molecule in electronic ground and excited states, also considering the influence of the environment ($CH_2Cl_2$, $CH_3OH$, and DMSO) by the Polarizable Continuum Model (PCM). The results of the simulations explained the effect of the solvent polarity and the specific interactions between the dye and the solvent.

## 6. Conclusions

A description of aromaticity is essential when posing questions about the genesis of the strengthening of hydrogen bonding in *quasi*-aromatic systems. By comparing two types of compounds (with RAHB (*Schiff* bases) and without RAHB (*Mannich* bases)), significant differences between these groups of compounds have been demonstrated. These studies have provided tools helpful in the analysis of compounds with RAHB, which is the dependence of the hydrogen bond energy on the HOMA aromaticity index of the *chelate* chain ($\Delta E_{HB} = f(HOMA(ch))$). The performed studies have stated that the *resonance* strengthens the intramolecular hydrogen bonding by ca. 30%.

A comparison of the results of the theoretical studies of the discussed compounds characterized by the presence of RAHB (dipyrromethene) and CAHB (dipyrromethane), revealed that, in terms of the global energy minimum, the charge strengthens the hydrogen bond more than the resonance. However, strengthening by the resonance in the intramolecular hydrogen bond at the transition state is comparable to strengthening by means of the charge.

Studies on hydrogen bonding in 2-hydroxyaryl amides by the infrared, UV-Vis, NMR, and quantum mechanical methods have proved that coupling between the amide group and phenol moiety is quite weak due to the competitiveness of two resonances—in the *chelate* chain and in the amide group. This effect causes quite easy disruption of a strong intramolecular hydrogen bond by the steric effect.

The combination of strong fluorescent BODIPY dyes with tautomeric sensitivity substituents (with the *ortho*-hydroxyaryl aldimine or diketone) has been designed. It turned out that modifying the interaction between the substituent and the core of the BODIPY dye allows one to govern the spectroscopic parameters (bathochromic and hypsochromic shifts), depending on the conformational state of the molecule. A significant effect of the specific interactions of the BODIPY dye with the proton-donor or the proton-acceptor solvents on the spectroscopic parameters was demonstrated.

In sum, integrated studies conducted by experimental and computational techniques have enabled researchers to accurately describe the nature of molecular interactions and explain the impact of various factors (steric effect, tautomeric and conformational equilibrium) on the physicochemical properties of compounds with intramolecular hydrogen bonding.

**Author Contributions:** Conceptualization, A.F.; Methodology, all authors; Software, all authors; Validation, all authors; Formal Analysis, all authors; Investigation, all authors; Resources, all authors; Data Curation, all authors; Writing-Original Draft Preparation, all authors; Writing-Review & Editing, all authors; Visualization, all authors; Supervision, A.F.; Project Administration, all authors; Funding Acquisition, all authors".

**Funding:** This research was funded by National Science Centre of Poland-grant number [UMO-2016/22/M/ST4/00226] and Russian Science Foundation - grant number [No. 18-13-00050].

**Acknowledgments:** This work has been supported by the National Science Centre of Poland (NCN, grant No. UMO-2016/22/M/ST4/00226) and Russian Science Foundation (RFBR, grant No. 18-13-00050).

**Conflicts of Interest:** The authors declare no conflicts of interest.

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
