# Peer review of "Intramolecular Hydrogen Bonds in Selected Aromatic Compounds: Recent Developments"

_catalysts, doi:10.3390/catal9110909_

Round 1

Reviewer 1 Report

In this manuscript, Filarowski and co-workers review the fundamental aspects of intramolecular hydrogen bonding (HBs) systems within aromatic compounds. HBs play important roles in enzyme, catalysts, and materials science. The topics compiled herein include the intramolecular systems of ortho-hydroxy benzimines, BODIPY-type molecules, and tautomeri keto-enol aromatic systems. IR, Raman, IINS as well as computational methods are useful for analyzing the structures of HBs. Interestingly, the photophysical properties of BODIPY type compounds with intramolecular HB systems are very sensitive toward the solvated media (i.e, aprotic or protic). Overall, the manuscript is concise and precise, and the concept is very clear. Therefore, if the authors fully consider and address the following points, I would support the publication of this work in Catalysts.

The data from Ref 54 are shown in Figures (e.g., Figure 13 and 14), but the Ref 54 does not seem to be published yet. If the paper has been published, the authors should add doi at the reference section. What is NQR? It would be more reader-friendly to define the abbreviation in the manuscript. The resolution of the Figure 16 is not good.

Author Response

Dear Reviewer,

we are grateful for the reviewing of our manuscript.

Reviewer 1.

In this manuscript, Filarowski and co-workers review the fundamental aspects of intramolecular hydrogen bonding (HBs) systems within aromatic compounds. HBs play important roles in enzyme, catalysts, and materials science. The topics compiled herein include the intramolecular systems of ortho-hydroxy benzimines, BODIPY-type molecules, and tautomeri keto-enol aromatic systems. IR, Raman, IINS as well as computational methods are useful for analyzing the structures of HBs. Interestingly, the photophysical properties of BODIPY type compounds with intramolecular HB systems are very sensitive toward the solvated media (i.e, aprotic or protic). Overall, the manuscript is concise and precise, and the concept is very clear. Therefore, if the authors fully consider and address the following points, I would support the publication of this work in Catalysts.

The data from Ref 54 are shown in Figures (e.g., Figure 13 and 14), but the Ref 54 does not seem to be published yet. If the paper has been published, the authors should add doi at the reference section. What is NQR? It would be more reader-friendly to define the abbreviation in the manuscript. The resolution of the Figure 16 is not good.

Answers to Reviewer 1.

Unfortunately, we haven’t got doi yet.

On page 7 we have added the decoded abbreviation of NQR. “4. The phase transition in one of the polymorphs was found out by the DSC method, which was confirmed by 35Cl NQR (Nuclear Quadrupole Resonance) study (Figure 14).”

Figure 16 has been enlarged and the spectra are more clearly visible now.

Sincerely,

Aleksander Filarowski

Reviewer 2 Report

Pr Filarowski and co-workers have submitted for publication in Catalysts a review, which presents the state of the arts in the field of hydrogen bonding for various series such as Schiff bases, Mannich bases ketones, dipyyrins, and amides, well-known species for their ability to form chelates through hydrogen bonding.

The introduction section clearly presents the objectives and particularly the fundamental role played by the H bonds and networks on the properties of catalysts in chemistry and biochemistry. The manuscript is divided in four main parts, each one related to the different families of derivatives described in this study, and it is well supported by a current bibliography section. The first part concerns the resonance assistance and charge assistance on the hydrogen bond properties in Schiff and Mannich bases and for dipyrromethane and dippyrromethane as well. The second part concerns quasi-aromatic cyclically arranged H bond systems and mainly summarizes recent works developed by the authors that have been published in the journal N.J.Chem. (2018) and. This section could deserve to be shortened. Equilibrium, which may exist between intra and inter molecular hydrogen bonds in benzamide derivatives is discussed in a third part. The latter part displays recent studies on the effect of substituents and solvatochromism for a series of the BODIPY chromophores.

This review, carefully presented and structured, is supported by a bibliographic section that includes 88 references, and mostly recent references. In this paper, the authors demonstrate the importance of the combination of experimental and computational studies. Such an approach is interesting as it allows to getting a better understanding of the mechanism of formation of intramolecular H bonds and further control hydrogen transfer, mechanisms that are important particularly in the field of catalysis. Besides, this review could interest chemists working in others fields such as for instance molecular materials sciences. This article would deserve to be published after minor revisions.

Figure 18: the characters and particularly the negative charges onto the boron atoms are too small.

Author Response

 Dear Reviewer

We are grateful for the reviewing of our manuscript.

Reviewer 2.

Pr Filarowski and co-workers have submitted for publication in Catalysts a review, which presents the state of the arts in the field of hydrogen bonding for various series such as Schiff bases, Mannich bases ketones, dipyyrins, and amides, well-known species for their ability to form chelates through hydrogen bonding.

The introduction section clearly presents the objectives and particularly the fundamental role played by the H bonds and networks on the properties of catalysts in chemistry and biochemistry. The manuscript is divided in four main parts, each one related to the different families of derivatives described in this study, and it is well supported by a current bibliography section. The first part concerns the resonance assistance and charge assistance on the hydrogen bond properties in Schiff and Mannich bases and for dipyrromethane and dippyrromethane as well. The second part concerns quasi-aromatic cyclically arranged H bond systems and mainly summarizes recent works developed by the authors that have been published in the journal N.J.Chem. (2018) and. This section could deserve to be shortened. Equilibrium, which may exist between intra and inter molecular hydrogen bonds in benzamide derivatives is discussed in a third part. The latter part displays recent studies on the effect of substituents and solvatochromism for a series of the BODIPY chromophores.

This review, carefully presented and structured, is supported by a bibliographic section that includes 88 references, and mostly recent references. In this paper, the authors demonstrate the importance of the combination of experimental and computational studies. Such an approach is interesting as it allows to getting a better understanding of the mechanism of formation of intramolecular H bonds and further control hydrogen transfer, mechanisms that are important particularly in the field of catalysis. Besides, this review could interest chemists working in others fields such as for instance molecular materials sciences. This article would deserve to be published after minor revisions.

Figure 18: the characters and particularly the negative charges onto the boron atoms are too small.

Answers to Reviewer 2.

According to the Referee suggestion we have reduced the part of the text which describes the results from our paper in NJC.

The text below has been removed from the manuscript and, consequently, the numbering of the figures has been changed further in the text and marked green.

The calculations confirmed that in the experimental spectra one should expect the presence of the asymmetric and symmetric stretching modes of the hydrogen bridge (ns(asym) and ns(sym)), as well as the single bending mode (nβ), Figure 11.

Figure 11. The scheme of vibrations of the hydrogen bridge in the tris-hydroxy ketoimines. Adopted and reprinted with permission from Ref. 46. Copyright 2018 The Royal Society of Chemistry.

Sincerely,

Aleksander Filarowski

Reviewer 3 Report

The article gives an extensive understanding of hydrogen bonding in quasi aromatic compounds. Although their authors have done a great job in quoting the latest development in the field, I find that it is not suitable for publication in this journal. It is certainly publishable in other more related organic journals. In addition, certain English sentences and words are not places correctly. 

There is an extensive quoting of other work done but it lacks the originality of thoughts and there has been close to nil opinions presented.  I suggest, rewriting it and submitting it in another journal. 

Author Response

Dear Reviewer

We are grateful for the reviewing of our manuscript.

Below the answers to the Reviewer are given.

Reviewer 3.

The article gives an extensive understanding of hydrogen bonding in quasi aromatic compounds. Although their authors have done a great job in quoting the latest development in the field, I find that it is not suitable for publication in this journal. It is certainly publishable in other more related organic journals. In addition, certain English sentences and words are not places correctly.

There is an extensive quoting of other work done but it lacks the originality of thoughts and there has been close to nil opinions  for

Answers to Reviewer 3.

As for the first remark, we would like to cite the fragments from the reviews of our paper by Reviewers 1, 2 and 4, who definitely consider our paper suitable for the publishing in this journal (the text below). We believe that the competent Reviewers 1, 2 and 4 adequately assess our paper. Also, we can’t understand what the Referee means pointing out “other work”? This manuscript is a review where we dwell on the selection of our papers published in the renowned journals, MDPI in that number. Our approach is clearly expressed by Referee 2. “This review, carefully presented and structured, is supported by a bibliographic section that includes 88 references, and mostly recent references. In this paper, the authors demonstrate the importance of the combination of experimental and computational studies. Such an approach is interesting as it allows to getting a better understanding of the mechanism of formation of intramolecular H bonds and further control hydrogen transfer, mechanisms that are important particularly in the field of catalysis.

The authors and the English service carefully scanned the text and necessary corrections have been introduced (marked in green).

Reviewer 1.

In this manuscript, Filarowski and co-workers review the fundamental aspects of intramolecular hydrogen bonding (HBs) systems within aromatic compounds. HBs play important role in enzyme, catalysts, and materials science.

Overall, the manuscript is concise and precise, and the concept is very clear. Therefore, if the authors fully consider and address the following points, I would support the publication of this work in Catalysts.

Reviewer 2.

Pr Filarowski and co-workers have submitted for publication in Catalysts a review, which presents the state of the arts in the field of hydrogen bonding for various series such as Schiff bases, Mannich bases ketones, dipyyrins, and amides, well-known species for their ability to form chelates through hydrogen bonding.

The introduction section clearly presents the objectives and particularly the fundamental role played by the H bonds and networks on the properties of catalysts in chemistry and biochemistry.

Reviewer 4.

The present manuscript reviews the area of chelate formations through intramolecular hydrogen bonding process in some aromatic compounds. Since chelation is an important factor in catalytic processes, the topic is of interest to the readership of this journal. In general I find the manuscript well-balanced and informative.

Sincerely,

Aleksander Filarowski

Reviewer 4 Report

The present manuscript reviews the area of chelate formations through intramolecular hydrogen bonding process in some aromatic compounds. Since chelation is an important factor in catalytic processes, the topic is of interest to the readership of this journal. In general I find the manuscript well-balanced and informative. Therefore, I do recommend acceptance.

My suggestion is to revise figure 21 for better readability. 

With this minor revision, I recommend acceptance.

Author Response

Dear Reviewer

We are grateful for the reviewing of our manuscript.

Reviewer 4.

The present manuscript reviews the area of chelate formations through intramolecular hydrogen bonding process in some aromatic compounds. Since chelation is an important factor in catalytic processes, the topic is of interest to the readership of this journal. In general I find the manuscript well-balanced and informative. Therefore, I do recommend acceptance.

My suggestion is to revise figure 21 for better readability.

With this minor revision, I recommend acceptance.

Answer to Reviewer 4.

The spectra on page 21 have been enlarged and now they are more clearly seen.

Sincerely,

Aleksander Filarowski